# A receptor-like kinase mutant with absent endodermal diffusion barrier displays selective nutrient homeostasis defects

Alexandre Pfister[1], Marie Barberon[1], Julien Alassimone[1†],
Lothar Kalmbach[1], Yuree Lee[1‡], Joop EM Vermeer[1], Misako Yamazaki[1§],
Guowei Li[2], Christophe Maurel[2], Junpei Takano[3], Takehiro Kamiya[4¶],
David E Salt[4], Daniele Roppolo[1#], Niko Geldner[1*]

[1]Department of Plant Molecular Biology, University of Lausanne, Lausanne, Switzerland; [2]Biochimie et Physiologie Moléculaire des Plantes, CNRS/INRA/SupAgro/Université Montpellier, Montpellier, France; [3]Research Faculty of Agriculture, Hokkaido University, Hokkaido, Japan; [4]Institute of Biological and Environmental Sciences, University of Aberdeen, Aberdeen, United Kingdom

**\*For correspondence:** niko.geldner@unil.ch

**Present address:** [†]Department of Biology, Stanford University, Stanford, United States; [‡]Center for Plant Senescence and Life History, Institute for Basic Science, Daegu, Republic of Korea; [§]Institute of Evolutionary Biology and Environmental Studies, University of Zürich, Zürich, Switzerland; [¶]Graduate School of Agricultural and Life Sciences, University of Tokyo, Tokyo, Japan; [#]Institute of Plant Sciences, University of Bern, Bern, Switzerland

**Competing interests:** The authors declare that no competing interests exist.

**Abstract** The endodermis represents the main barrier to extracellular diffusion in plant roots, and it is central to current models of plant nutrient uptake. Despite this, little is known about the genes setting up this endodermal barrier. In this study, we report the identification and characterization of a strong barrier mutant, *schengen3* (*sgn3*). We observe a surprising ability of the mutant to maintain nutrient homeostasis, but demonstrate a major defect in maintaining sufficient levels of the macronutrient potassium. We show that *SGN3/GASSHO1* is a receptor-like kinase that is necessary for localizing CASPARIAN STRIP DOMAIN PROTEINS (CASPs)—major players of endodermal differentiation—into an uninterrupted, ring-like domain. SGN3 appears to localize into a broader band, embedding growing CASP microdomains. The discovery of *SGN3* strongly advances our ability to interrogate mechanisms of plant nutrient homeostasis and provides a novel actor for localized microdomain formation at the endodermal plasma membrane.

## Introduction

The plant root is a highly selective filter that forages the soil environment for nutrients and water. Its function has been likened to that of an 'inverted gut', displaying a very similar dual role in uptake and protection (*Waisel et al., 2002*). Selectivity of roots is thought to crucially depend on the endodermis, an inner cell layer that surrounds the central vascular strand of the root and represents the main para-cellular (apoplastic) transport barrier in young roots (*Geldner, 2013*). This apoplastic diffusion barrier is set up by the 'Casparian strips', lignin-like, hydrophobic impregnations of the primary cell wall that form a supracellular net-like structure around the central vasculature. In analogy to the tight junction of gut epithelia, this net-like arrangement provides a seal of the extracellular space, while still allowing for nutrient and water transport across outer and inner plasma membrane surfaces. The Casparian strip-bearing endodermis therefore represents an independent development of a polarized epithelium, evolved in the context of multi-cellular organisms with wall-bearing cells.

Later during endodermal development, hydrophobic (cork-like) suberin lamellae form all around the endodermal surface. This protective suberization should eventually abolish direct transport across the endodermal plasma membrane, forcing symplastic passage through plasmodesmata from outer cell layers. Despite being a central, conserved feature of higher plants, nothing was known until recently about the factors that build and position the Casparian strip. This led to the frustrating situation of not

**eLife digest** Plant roots forage in the soil for minerals and water, but they must also provide a barrier that stops these nutrients leaking back out of the plant and stops microbes invading and causing disease. The endodermis—an inner layer of cells that surrounds the veins that run along the middle of a root—acts as such a barrier in young roots.

Polymers that repel water are deposited between the cells in the roots of almost all vascular plants—which include ferns, conifers, and flowering plants—to form a band around the endodermis called the 'Casparian strip'. This strip seals off the young roots and stops water moving through the gaps between plant cells, but still allows minerals, nutrients, and water to be transported through the root cells and into the plant. However, the importance of this structure has yet to be tested due to the lack of mutant plants without a Casparian strip.

Pfister et al. now report that deleting the gene that encodes a protein called SCHENGEN3 in the model plant *Arabidopsis thaliana* causes the Casparian strip to be interrupted by irregularly sized holes. This protein is normally found at high levels in the root endodermis, where it is embedded into the cell membranes. Pfister et al. also showed that without the SCHENGEN3 protein, other proteins called CASPs—that normally mark out a stripe around the root cells where the Casparian strip will form—only accumulated in discontinuous patches. Further experiments revealed that deleting the gene for SCHENGEN3 does not cause general problems in delivering the CASP proteins to the cell membrane; instead, it specifically stops the CASP proteins from forming a single, uninterrupted stripe.

Unexpectedly, disrupting the Casparian strip did not appear to hinder many of the functions of a root. The mutant plants could still take up water and nutrients, and the leaves of mutant plants had normal levels of many essential minerals—with the exception of potassium. The level of this mineral was much lower in mutant plants without the SCHENGEN3 protein. Pfister et al. suggest that in plants that lack an intact Casparian strip, potassium is continuously leaked from the root into the soil.

These findings reveal that in *Arabidopsis*, at least, the Casparian strip might not be as important as once thought for helping the plant to take up and accumulate water and nutrients. Further work is now needed to uncover the as yet unknown backup systems that might be able to compensate for the loss of this structure.

being able to test the many supposed roles of the endodermis and its Casparian strip, because of an absence of specific mutants. Recent papers established that a family of previously undescribed four-transmembrane-span proteins, called CASPs, forms a central, ring-like membrane domain in the endodermis, called the Casparian strip membrane domain (CSD). This domain acts as a lateral diffusion barrier and separates the endodermal plasma membrane into two distinct polar domains (*Roppolo et al., 2011*; *Alassimone et al., 2010*). Upon localization, the CASPs show extreme stability and a lack of endocytosis or lateral diffusion. Moreover, CASPs show extensive pair-wise interactions and associate strongly with cell walls. Multiple knock-outs of *CASP* family members display strongly disorganized formation of Casparian strips, demonstrating their functional importance. It is thought that the CASPs act as scaffolds that spatially organize cell wall biosynthetic enzymes. *CASP1* was shown to determine the subcellular localization of a secreted peroxidase, *PER64* (*Lee et al., 2013*). In current models, the role of CASPs is to assemble a specific NADPH oxidase with peroxidases, leading to oxidation of mono-lignols and local polymerization of lignin. Lignin formation in vivo must implicate numerous other cell wall proteins and ENHANCED SUBERIN 1 (ESB1), a dirigent-like protein is a cell wall protein that has also been demonstrated to be crucial for the correct formation of Casparian strips (*Hosmani et al., 2013*).

These recent breakthroughs held the promise that we might finally be able to investigate the functional relevance of the Casparian strip through analysis of the different mutants obtained. This however, turned out to be more difficult than expected. Multiple mutants of the *CASP* family display interrupted Casparian strips in the beginning, but rather quickly deposit more cell wall material in a delocalized fashion, eventually sealing the apoplast. A very similar phenotype occurs in the *esb1* mutant (*Hosmani et al., 2013*). Moreover, both mutants display an earlier and stronger production of

suberin lamellae (*esb1* was named based on this phenomenon). This makes it difficult to use these mutants for assessing the importance of Casparian strips, since the partial defect on Casparian strip formation will always be confounded with the effect of ectopic/enhanced production of suberin in the same cells. In the *rbohf* mutant, the Casparian strip defect is only partial and restricted to the young part of the root. Moreover, *RBOHF* has roles in many different aspects of plant development and it would be hard to distinguish what phenotype is directly caused by defects in Casparian strip formation (*Lee et al., 2013*).

Here, we report the discovery of a novel Casparian strip mutant, *schengen3* (*sgn3*). *SGN3* encodes a receptor-like kinase with strong expression in the root endodermis. The SGN3 protein accumulates in the plasma membrane in a broad band within which the CSD forms. In its absence, only discontinuous patches of CASPs are observed. *SGN3* loss-of-function leads to the strongest defects in Casparian strips known to date, with no indication of a compensatory upregulation of suberin, as seen for other mutants (*Hosmani et al., 2013*). Our analysis reveals a surprising capacity of the mutant to maintain homeostatic control in the absence of the major root diffusion barrier and challenges views according to which the root should lose its ability for selective nutrient uptake, because of a generalized, non-selective bypass of nutrients into the vasculature.

## Results

### An endodermis-enriched receptor-like kinase is necessary for Casparian strip formation

In a forward genetic, GUS-based screen for endodermal barrier mutants (*Lee et al., 2013*) (Alassimone et al., *unpublished*), we discovered one mutant, *sgn3* that displayed a dramatic defect in endodermal barrier formation, as visualized by penetration of the apoplastic tracer propidium iodide (PI) into the stele along the entire length of the root. This resembled a phenotype of T-DNA insertion lines in an endodermis-enriched gene that we had analyzed in parallel (*Figure 1A*, *Figure 1—figure supplement 1A*) (*Birnbaum et al., 2003*; *Brady et al., 2007*). Complementation analysis and sequencing confirmed that the causal mutation in *sgn3-2* was an early stop codon in the open reading frame of *At4g20140*. *SGN3* encodes for a leucine-rich-repeat receptor-like kinase (LRR-RLK) of subfamily XI that had previously been described as *GASSHO1* (*GSO1*) (*Figure 1B*, *Figure 1—figure supplement 2*, *Figure 1—source data 1*) (*Tsuwamoto et al., 2008*). In the previous publication, this LRR-RLK was investigated because of embryonic expression of a putative ortholog in *Brassica napus*, but no single mutant phenotype was found. A phenotype was only observed when *gso1* (*sgn3*) was combined with a mutant of its closest homolog, *At5g44700* (named *GSO2*). This *gso1 gso2* double mutant displayed a protodermal phenotype, characterized by a defective cuticle and fusion of cotyledons (*Tsuwamoto et al., 2008*). True leaves did not differ from wild type. In contrast to *SGN3/GSO1*, *GSO2* was shown to be expressed only in shoots, but not in roots, explaining the exclusive root phenotype in the *sgn3* single mutant that we report here and that had escaped detection earlier on. In order to investigate the cause of this phenotype, we investigated formation of the Casparian strip and found that it is still formed and correctly positioned, but that it is repeatedly interrupted, forming irregularly-sized 'holes' of several micrometer in length (*Figure 1C*, *Figure 1—figure supplement 1B*). Suberin lamellae formation, by contrast, appeared at a position and in a manner very similar to wild type (*Figure 1D,E*, *Figure 1—figure supplement 1C,D*).

### The Casparian strip membrane domain and lateral diffusion barrier is defective in the *sgn3* receptor-like kinase mutant

We asked whether the Casparian strip cell wall defect is caused by a defect in the formation of the CSD, or whether the function of *SGN3* lies downstream of the *CASPs*, as has been demonstrated for the NADPH oxidase *SGN4/RBOHF* (*Lee et al., 2013*). In the *sgn3* background, CASP1-GFP showed a pattern of interrupted patches that was strikingly similar to that of the Casparian strip itself. This suggests that it is the incorrect localization of the CASPs that causes the Casparian strip defects of the mutant (*Figure 1F*). Interestingly, in a novel forward genetic screen in which we scored directly for mutants with altered CASP1-GFP localization (Kalmbach et al., *unpublished*), we found 14 novel alleles of *sgn3* (*Figure 1B*, *Figure 1—figure supplement 2*, *Figure 1—source data 1*). This suggests that *SGN3* is one of the major non-redundant genes necessary for CASP localization in the *Arabidopsis* genome. We could show that not only CASP1-GFP, but all four other CASPs (CASP2-5) display identical defects in *sgn3* (*Figure 1—figure supplement 1E*).

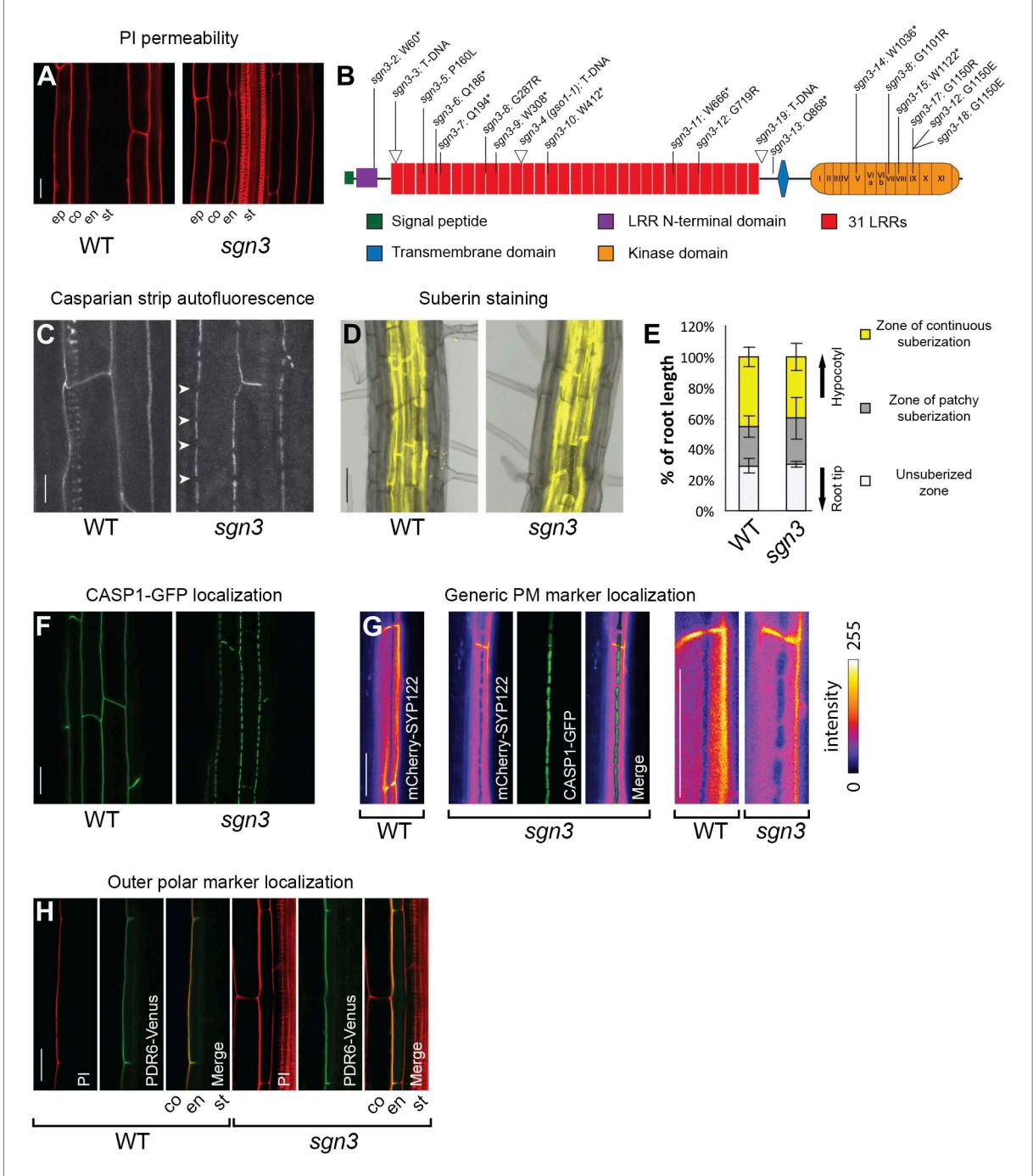

**Figure 1**. *SGN3* receptor-like kinase is important to establish a functional endodermal barrier. (**A**) Lack of endodermal diffusion barrier in *sgn3-3* visualized by presence of propidium iodide (PI) in stele. (**B**) Diagram of the SGN3 protein showing the different domains, T-DNA insertion lines (indicated with triangles) and the ethyl methanesulfonate (EMS)-induced mutations (see also *Figure 1—figure supplement 2*). (**C**) Surface view of Casparian strip, visualized by autofluorescence after clearing. Note discontinuous lignin deposition in *sgn3-3*. Pictures are maximum projections of confocal z-stacks. Arrowheads indicate discontinuities in *sgn3*. Spiral-like signal in WT is from deeper-lying xylem vessel. (**D**) Fluorol yellow staining of suberin lamellae deposition in *sgn3-3* and WT. Pictures are overlays of transmitted light image (gray) with fluorescent signal from suberin dye (yellow). (**E**) Occurrence of suberin deposition along the root is not altered in *sgn3-3*. Suberin lamellae deposition was quantified considering three different zones: non-suberized zone, zone of patchy suberization, and zone of continuous suberization (n = 5, one representative experiment presented). (**F**) Surface view of CSD network visualized with CASP1-GFP expressed under *CASP1* promoter showing the net-like structure with discontinuities in *sgn3-3*. Projections as in **C**. (**G**) Absence of a lateral diffusion barrier in *sgn3-3* visualized with plasma membrane marker line *CASP1::mCherry-SYP122* (intensity color coded). *Figure 1. Continued on next page*

*Figure 1. Continued*

Confocal pictures were taken at the surface of an endodermal cell. Note the mutually exclusive localization with the CSD marker CASP1-GFP (green). Two right images are magnification of the two leftmost images. (**H**) Localization of the outer marker PDR6-Venus expressed under *CASP1* promoter is still polar in *sgn3-3*. Pictures are median longitudinal sections of endodermal cells. Scale bars: **A**, **C**, **F**, **G**, **H** = 20 µm; **D** = 50 µm. ep, epidermis; co, cortex; en, endodermis; st, stele; LRR, Leucine-rich repeat.

The following source data and figure supplements are available for figure 1:

**Source data 1**. Detail of *SGN3* T-DNA and EMS mutants.

**Figure supplement 1**. Both Casparian strip domain and Casparian strip but not the suberin are affected in *sgn3*.

**Figure supplement 2**. Diagram of *SGN3* genomic DNA with T-DNA and EMS mutants.

An interrupted CSD should abolish the lateral diffusion barrier in the plasma membrane of endodermal cells that we had previously described (*Alassimone et al., 2010*). Indeed, a generic plasma membrane protein that is excluded from the CSD in wild type (*Alassimone et al., 2010*) (*Figure 1G*) shows a pattern of exclusion in the mutant that represents a perfect negative image to the CASP1-GFP islands (*Figure 1G*). This illustrates that the lateral diffusion barrier between inner and outer plasma membrane domain is absent in *sgn3* mutants, but that the remaining CSD islands are still able to exclude other proteins. To our surprise, this defect did not affect the polar localization of PDR6, a transporter that localizes to the outer polar domain in the endodermis (*Figure 1H*). This indicates that a strict polarity in the endodermis can be maintained in the absence of a diffusion barrier, further supporting the notion that polarity of transmembrane proteins in plants might simply be maintained by generally low lateral diffusion, high rates of endocytosis, or both (*Geldner, 2009*).

## CASP stability and cell wall attachment remain intact in the *sgn3* mutant

We had shown previously that during endodermal differentiation, CASP1-GFP changes from a protein that displays endocytic cycling and lateral diffusion to one that is highly stable and immobile in the CSD (*Roppolo et al., 2011*). In this process, CASP1-GFP localization passes through a 'string of pearls' stage, in which individual islands or 'patches' of CASP1-GFP are aligned in a band before eventually fusing (*Figure 2A*). Comparison of 3D-time-lapse observations of endodermal differentiation revealed that *sgn3* is not able to progress from this 'string of pearls' stage, but only undergoes partial fusion of individual islands, leading to CASP1-GFP patches of heterogenous size (*Figure 2A*). We further found by fluorescence recovery after photobleaching (FRAP) that these CASP1-GFP islands have similar stability and turnover as wild type (*Figure 2B*). Another feature of the CSD is its very strong adherence to the cell wall, leading to the long-described phenomenon of 'band plasmolysis' in endodermal cells (*Krömer, 1903*; *Behrisch, 1926*; *Alassimone et al., 2010*). The inability of the CASP1-GFP ring to retract from the Casparian strip upon plasmolysis (*Figure 2—figure supplement 1*) allow for easy visualization of this phenomenon. This adhesion leads to greatly flattened, often fenestrated protoplasts, that looked very unlike the usual, rounded protoplasts of epidermal cells (*Alassimone et al., 2010*). *sgn3* showed the very same inability of CASP1-GFP to retract from the strips, demonstrating that plasma membrane-to-cell wall attachment remains intact in the mutant (*Figure 2—figure supplement 1*). We conclude that *sgn3* does not cause a general defect in CASP1-GFP trafficking, polymerization or cell wall attachment, but rather a specific defect in the progression of already stable CASP islands towards a contiguous band.

## SGN3 localizes to the plasma membrane and is expressed early during endodermal differentiation

We then generated lines of tagged SGN3 variants in order to investigate its expression and subcellular localization. Only a C-terminal mVenus fusion with a 9.4-kb genomic fragment containing intron, 5'UTR, and the upstream neighboring gene, provided full complementation of the mutant phenotype, indicating functionality of the protein fusion and of its regulatory sequences (*Figure 2—figure supplement 2A,B*).

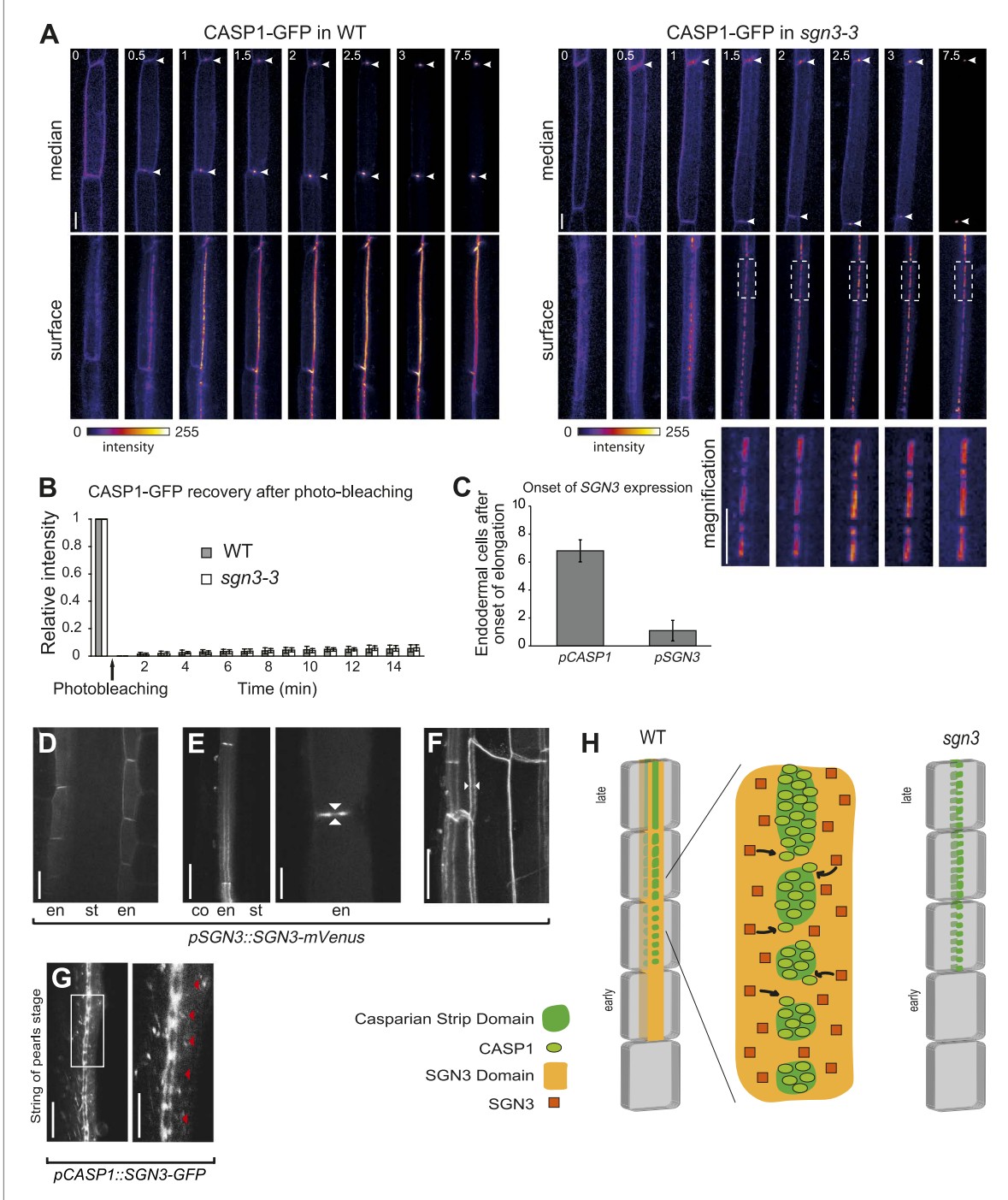

**Figure 2**. SGN3 localizes on both sides of the Casparian strip domain and is important for the CASP1-GFP patches to fuse into a contiguous band.
(**A**) 3D-confocal time lapse imaging of *CASP1::CASP1-GFP* in Col-0 and *sgn3-3* background reveals problems with progression of CASP1-GFP localization in *sgn3-3*. Images show median and surface image of endodermal cells of the same root. Arrowhead indicates CSD, dotted box in *sgn3-3* shows immobility of CSD islands. Time in hours. (**B**) Fluorescence recovery after photobleaching (FRAP) with *pCASP1::CASP1–GFP* in WT (gray) and *sgn3-3* (white). The fifth cell after the onset of *CASP1* expression was used. Similar very low recoveries are observed in both genotypes after 15 min. n = 8 independent assays for each series, error bars = s.d. (**C**) *SGN3* is expressed before the onset of *CASP1* expression in the endodermis. Quantification was done using *pSGN3::SGN3-mVenus* and *pCASP1::CASP1-GFP*. The onset of expression corresponds to the first cell in the endodermis file with a clear fluorescent signal not visible in WT. n = 10. Error bars = s.d. (**D**) Localization of SGN3-mVenus under its own promoter in elongating endodermal cells. (**E**) In differentiated endodermal cells, SGN3-mVenus accumulates in the transversal and anticlinal sides of the plasma membrane, but is depleted from the CSD. Left panel shows a transition from a median (top) to a surface (bottom) view of an endodermal cell. Right panel shows a close-up of a median
*Figure 2. Continued on next page*

*Figure 2. Continued*

view with SGN3 surrounding the CSD. Arrowheads indicate the CSD. (**F**) Maximum projection of a z-stack showing the localization of SGN3 on both sides of the CSD. Note the CSD surrounded by two SGN3-mVenus 'lines' (white arrowheads). (**G**) Localization of SGN3-GFP under *CASP1* promoter in the 'string of pearls' stage. SGN3 surrounds individual CSD patches preceding their fusion (left panel). Close-up showing the CSD patches (red arrowhead) surrounded by SGN3-GFP (right panel). (**H**) Schematic of SGN3 putative mode of action. The SGN3 kinase might promote addition of new CASP units to already formed CASP microdomains. Scale bars: **A**, **G** (left panel), 10 μm; **D**, **E** (left panel), **F**, 20 μm; **E** (right panel), **G** (right panel), 5 μm. Ep, epidermis; co, cortex; en, endodermis; st, stele.

The following figure supplements are available for figure 2:

**Figure supplement 1**. Plasma membrane-to-cell wall attachment remains intact in *sgn3* upon plasmolysis.

**Figure supplement 2**. *SGN3* genomic construct and *sgn3* PI phenotype complementation.

**Figure supplement 3**. Schematic illustrating quantification of onset of expression along the root.

In transgenic lines of this construct, expression of SGN3 could be observed in endodermal cells shortly after the onset of elongation, preceding the onset of *CASP1* expression (*Figure 2C*, *Figure 2—figure supplement 3*). This is consistent with a function of SGN3 in establishing correct CASP1 localization. Initially, SGN3 could be found on all cell sides, but it then quickly accumulated in the transversal and anticlinal sides of the plasma membrane, yet became excluded from the CSD itself—a hitherto unknown subcellular localization pattern (*Figure 2D–F*). In 3D-reconstructions this leads to the appearance of a broader ring-like domain that flanks the more restricted, median CSD (*Figure 2F*). During the critical phase of CASP domain progression from isolated islands towards a fused ring—at which the *sgn3* phenotype becomes apparent–SGN3 is seen to surround individual CASP islands on all sides (*Figure 2G*). Such localization would fit with a role of SGN3 in promoting fusion of growing CASP islands, such that an uninterrupted band can be formed (*Figure 2H*).

## SGN3 is required for the enhanced suberin production of other Casparian strip-defective mutants

In order to establish the functional relationships between *SGN3* and other previously characterized genes, we combined *sgn3* with *casp1 casp3* double and *esb1* single mutants. ESB1 is an extracellular protein that localizes to the site of Casparian strip formation and whose absence causes a phenotype resembling that of *sgn3* insofar as both the Casparian strip itself and CASP1-GFP are localized in interrupted bands (*Hosmani et al., 2013*). In contrast to *sgn3*, however, *esb1* mutants additionally display increased, delocalized deposition of additional autofluorescent cell wall material as well as enhanced and earlier deposition of suberin (*Hosmani et al., 2013*). We found ESB1-mCherry in *sgn3* to be both present and correctly localized, that is, specifically accumulating at the remaining CASP1-GFP islands (*Figure 3A*). Thus, ESB1 does not require SGN3 for its accumulation and localization and the Casparian strip discontinuities in *esb1* and *sgn3* might be caused by independent mechanisms. Consistently, we found that *sgn3 esb1* double mutant formed less and/or smaller Casparian strip patches than each single mutant, indicating additivity of the two mutations in this common aspect of their phenotypes (*Figure 3B,C*). Intriguingly however, *sgn3 esb1* double mutants neither display an increased or delocalized deposition of autofluorescent cell wall material, nor an enhanced and earlier formation of suberin. This indicates a full epistasis of *sgn3* over *esb1* for this phenotype (*Figure 3B,D*). The same epistatic relationship was found between *sgn3* and *casp1 casp3* double mutants (*Figure 3B–D*). The enhanced and ectopic formation of autofluorescence and suberin is thought to be due to a feedback and cross-talk between cell wall components, connecting correct Casparian strip formation to the further deposition of lignin and suberin lamellae formation (*Hosmani et al., 2013*). The epistasis of *sgn3* suggests that the perception/signaling that leads to enhanced autofluorescence and suberin formation in *esb1* and *casp1 casp3* is mediated by—or at least requires—the SGN3 receptor-like kinase.

## sgn3 growth phenotypes are extremely sensitive to environmental conditions

Considering the strength of the barrier phenotype of *sgn3* and its inability to initiate potentially compensatory suberization, we were surprised to observe a rather mild growth phenotype of *sgn3* under

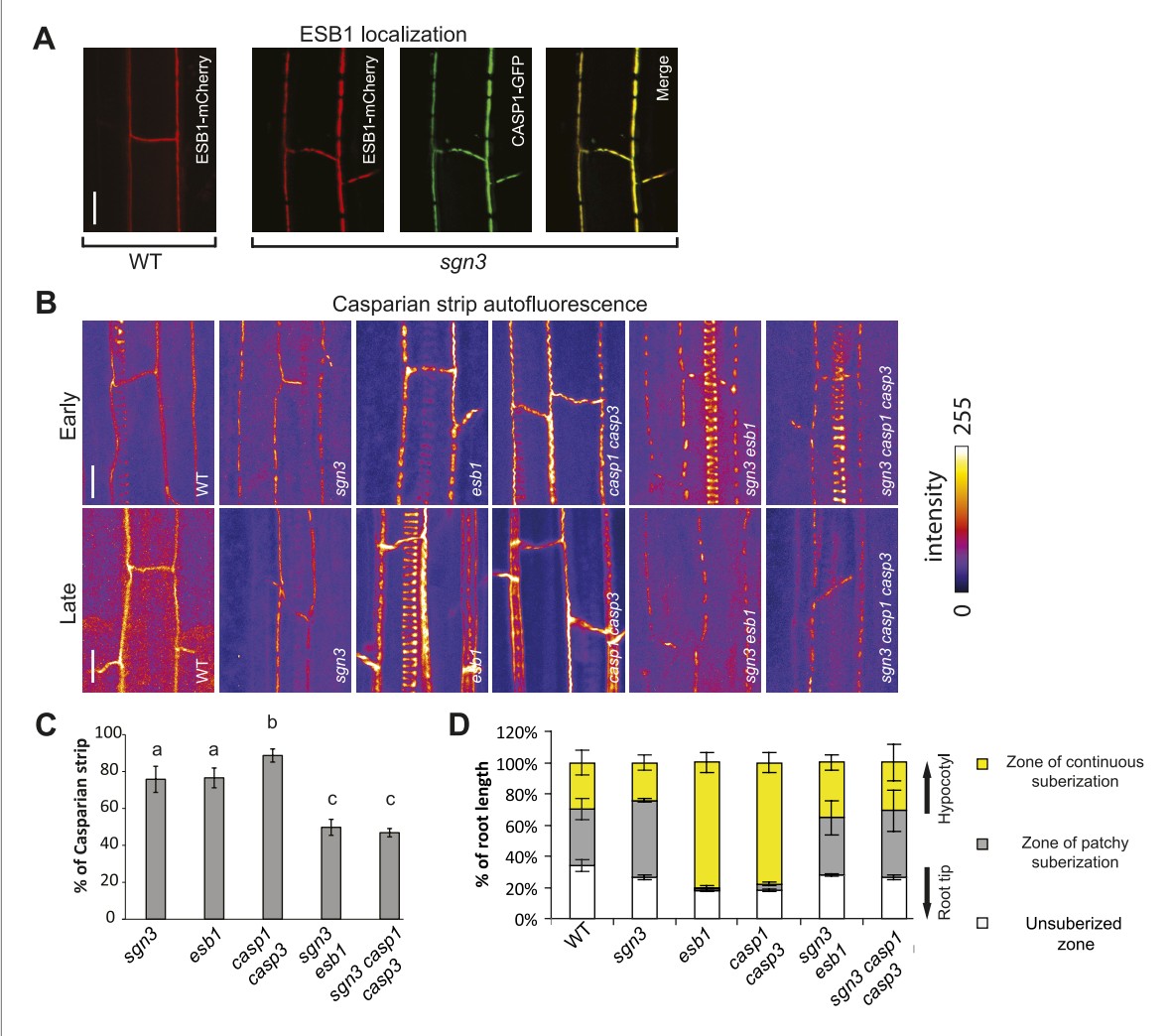

**Figure 3**. Relations between *sgn3* and molecular players in Casparian strip formation. (**A**) ESB1-mCherry localizes to the CASP1-GFP microdomains in *sgn3*. ESB1-mCherry and CASP1-GFP are expressed under their own promoters. Pictures are maximum projections of confocal z-stacks. (**B**) Surface view of Casparian strip, visualized by autofluorescence after clearing. Top and bottom panels correspond to the early and late stages of endodermal differentiation, 15 and 30 cells after onset of elongation, respectively. When *sgn3* is crossed to *casp1 casp3* or *esb1*, note the additive effect of these mutants concerning the CS patches (quantified in **C**). Note also the absence of enhance/ectopic deposition of lignin in *sgn3 esb1* and *sgn3 casp1 casp3*. Pictures are maximum projections as in **A**. Strong spiral-like signals are from protoxylem cells. (**C**) Quantification of Casparian strip presence shows an additive effect in *sgn3 esb1* and *sgn3 casp1 casp3*. Casparian strips were quantified as the percentage of cell wall showing autofluorescence along a line of Casparian strip signal of a given length. n = 10. (**D**) *sgn3* is epistatic to *esb1* and *casp1 casp3* for ectopic suberin deposition. 5-day-old seedlings were stained with Fluorol Yellow. Quantification was done as in **Figure 1E**. Bars represent the percentage for each zone along the root (n = 5, one representative experiment). **C**, Error bars = s.d., different letters indicate significant differences between genotypes, determined by analysis of variance (ANOVA) and Tukey test as *post hoc* analyses (p < 0.05). Scale bars: **A**, **B**, 10 μm. *sgn3* is *sgn3-3*, *esb1* is *esb1-1*, *casp1* is *casp1-1* and *casp3* is *casp3-1*.

certain conditions (**Figure 4A**, **Figure 4—figure supplement 1A**). Yet we noticed that *sgn3* growth is extremely sensitive to changes in environmental conditions that are non-stressful to wild type (**Figure 4A**, **Figure 4—figure supplement 1A**). We ensured that the endodermal barrier phenotype persisted in developed root systems of rosette stage plants both on soil and on hydroponics, thus excluding that the weak phenotype is caused by an eventual repair of the barrier in older plants (**Figure 4—figure supplement 1B**). A systematic variation of day length, temperature, and light-intensity revealed that growth differences to wild type are more pronounced at higher temperature, as well as under long-day conditions (**Figure 4B**, **Figure 4—figure supplement 1C**). In general, sensitivity of *sgn3* to growth conditions is such that we observed everything from a near absence of phenotypes to severely dwarfed

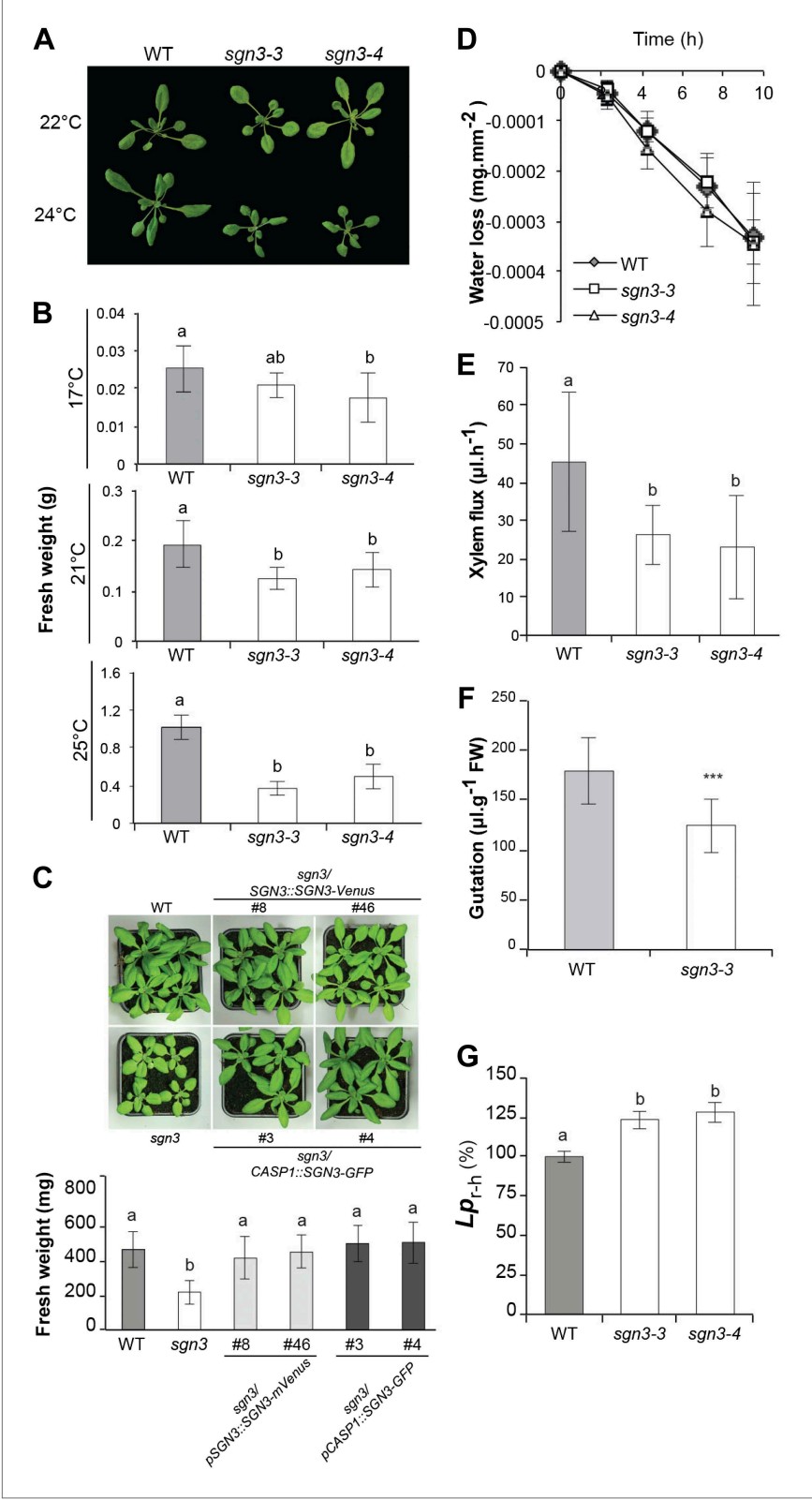

**Figure 4**. *sgn3* is sensitive to environmental conditions and displays an altered water transport and root pressure. (**A**) Phenotype of 3-week-old WT, *sgn3-3,* and *sgn3-4* plants grown at 22 or 24°C in long days. Representative pictures are presented. (**B**) Analysis of shoots fresh weight of WT, *sgn3-3,* and *sgn3-4* plants (n = 15) grown 3 weeks

*Figure 4. Continued on next page*

*Figure 4. Continued*

at different temperatures (17, 21, or 25°C). (**C**) Phenotype of 4-week-old WT, *sgn3-3*, *sgn3-3/pSGN3::SGN3-mVenus* (lines 8 and 46), and *sgn3-3/pCASP1::SGN3-GFP* (lines 3 and 4) grown at 24°C in long days. Representative pictures are presented. Fresh weight average from n > 8 plants. (**D**) Transpiration of WT, *sgn3-3*, and *sgn3-4* plants determined as water loss from 3-week-old plants. Error bars = s.d. (n = 10). (**E**) Root pressure analysis determined as the volume of xylem sap released in 30 min from decapitated WT, *sgn3-3*, and *sgn3-4* plants grown in short day condition (n = 7). (**F**) Guttation was collected from WT and *sgn3-3* plants grown for 6 weeks in short day conditions (n = 15). (**G**) Mean hydrostatic hydraulic conductivity of roots ($Lp_{r-h}$) from WT, *sgn3-3*, and *sgn3-4* plants. $Lp_{r-h}$ was measured during the daytime. Values correspond to means ± SD (n > 14). **B**, **C**, **E**, **F**, **G**. Error bars = s.d. For multiple comparison, different letters indicate significant differences between genotypes, determined by analysis of variance (ANOVA) and Tukey test as *post hoc* analyses; (**B**, p < 0.01) (**E**, **G**, p < 0.05). For single comparison in **F**, stars (⁎⁎⁎) indicate significant difference determined by Student test (p < 0.001).

The following figure supplement is available for figure 4:

**Figure supplement 1**. Impact of environmental conditions on growth of *sgn3* mutant.

plants that did not reproduce when grown in different, standard growth chambers that all well supported wild type growth. We wanted to ascertain that the observed growth phenotypes of *sgn3* are indeed caused by the Casparian strip defect in the roots and not by some hypothetical, non-redundant function of *SGN3* in shoots. We therefore expressed SGN3 under the control of the *CASP1* promoter that shows specific expression in differentiating root endodermis, with some expression in the endodermis of the lower hypocotyl and none detectable anywhere else in the plant. These *CASP1::SGN3-GFP sgn3* plants could fully rescue the growth phenotype of *sgn3* plants (**Figure 4C**), demonstrating that it is the root endodermal defect of *sgn3* that is causative for the overall growth defect of the mutant.

## *sgn3* mutants display normal transpiration but altered water transport and root pressure

The numerous, repeated Casparian strip discontinuities in *sgn3* must generate an apoplastic bypass for both water and solutes in the mutant root. Especially under conditions of high transpiration, this would be expected to lead to a strong, uncontrolled and possibly detrimental influx of elements into the transpiration stream. We observed that transpiration rate per surface area in *sgn3* mutants are very similar to wild type (**Figure 4D**) revealing that there is no compensatory downregulation of transpiration. Another predicted consequence of interrupted Casparian strips is an inability to build up root pressure, a process necessary for vascular transport of nutrients and water in the absence of transpiration and for transport into non-transpiring organs (**Wegner, 2014**). Root pressure build-up in the dead xylem vessels is thought to involve an osmotic gradient, generated by living neighbors, that draws water into the vessels. Pressure build-up, however, is thought to crucially depend on the presence of the Casparian strips as an apoplastic diffusion barrier, without which xylem pressure should quickly dissipate into the cortex. We indeed found that root pressure, as estimated by xylem sap exudation from hypocotyls after decapitation, was lower in the *sgn3* mutant (**Figure 4E**). *sgn3* also showed a decreased rate of leaf guttation, a process more indirectly dependent on root pressure, but that can be observed less invasively (**Figure 4F**). As a more direct measure of apoplastic water flow, we inserted excised roots in a pressure chamber and found a significant increase in root hydrostatic hydraulic conductivity ($Lp_{r-h}$) in *sgn3* mutants compared to wild type (**Figure 4G**). The defects of Casparian strip formation may allow an apoplastic bypass for water in pressurized *sgn3* roots thereby explaining its increased $Lp_{r-h}$. Our findings are the first genetic evidence that Casparian strips can be relevant for root pressure buildup and impose a hydraulic resistance; the magnitude of the observed effects nevertheless indicates that an intact Casparian strip is not an absolute requirement and that unmodified walls and/or suberized cells can generate a sufficiently high resistance to radial waterflow so that the positive root pressure can be maintained. *sgn3* might be very useful to dissect the mechanisms of root pressure formation and its role in nutrient transport in the future.

## Transcriptional profiles suggest a latent potassium deficiency in *sgn3* mutants

In order to identify some potential stress and/or compensatory responses in the *sgn3* mutant, we generated a general transcriptional profile of *sgn3* mutant rosettes, grown under conditions in

which wild type and *sgn3* were indistinguishable. Only little differences between *sgn3* and wild type were observed. Yet, all of the seven more highly expressed genes in *sgn3* were found to be also induced by potassium starvation, suggesting a latent, weak potassium deficiency stress under these conditions (*Figure 5A*, *Figure 5—source data 1*). Based on this indication, we tested well-established transcriptional read-outs of potassium deficiency in roots, the potassium influx transport proteins AKT1 and HAK5, and found that both were upregulated in *sgn3* (*Figure 5B,C*).

## An impaired endodermal barrier causes surprisingly specific defects in elemental homeostasis

In order to directly measure how *sgn3* root barrier defects affect elemental homeostasis in leaves, we undertook a number of elemental profiling experiments using inductively coupled plasma–mass spectrometry (ICP-MS) on rosette leaves in different laboratories under different growth conditions. Surprisingly, concentrations of most elements remained essentially unaltered in *sgn3*. Despite the dramatic apoplastic bypass in *sgn3* roots, the mutants managed to maintain wild type levels of sodium (Na), calcium (Ca), or boron (B), for example. Among the measured transition elements (Mo, Co, Mn, Zn, Fe, Cu, Zn) only zinc showed a significant decrease in some measurements (*Figure 5D*, *Figure 5— source data 2*). However, matching the transcriptional profiling results, we invariably found lower levels of potassium (K) in the mutant (ranging from 1.4–3.0 fold reduction). Inversely, levels of magnesium (Mg) and cesium (Cs) accumulated to higher levels than in wild type, ranging from 1.5–2.1 and 1.3- to 1.4-fold increase, respectively (*Figure 5D*, *Figure 5—source data 2*). These findings nicely corroborated the results of our microarrays, but left us wondering about the absence of potassium deficiency symptoms in *sgn3*, since the mutant phenotypes under most conditions are simply those of a reduction and delay in growth.

## The *sgn3* mutant is strongly hypersensitive to low potassium conditions

We then tested growth of *sgn3* plants on a nutrient-poor, gravel-like substrate, which we watered with a nutrient solution with or without potassium. Wild type plants coped well with conditions of low potassium nutrition, with most plants displaying no or only very weak chlorosis on older rosette leaves, indicative of potassium deficiency (*Figure 5E,F*). Consistently, wild type also maintained higher potassium concentration than *sgn3* under these conditions (*Figure 5—figure supplement 1*). The *sgn3* mutant in contrast, developed severe chlorosis of its rosette leaf margins, characteristic of strong potassium deficiency (*Marschner and Marschner, 2012*). *sgn3* therefore has an impaired capacity to accumulate sufficient potassium under conditions of low supply of this element.

## Discussion

### The *SGN3* receptor-like kinase is important for microdomain organization and enhanced suberization in the endodermis

The deposition of cell wall material with subcellular precision is crucial for the correct function of many plant cell types (*Roppolo and Geldner, 2012*). How this localized deposition is achieved is only rarely understood in any mechanistic detail, with the notable exception of wall deposition in metaxylem cells (*Oda and Fukuda, 2012*). Localized wall deposition should implicate the formation of microdomains in the plasma membrane that localize cell wall biosynthetic activity. Walls, in turn, signal back to the cell, and this process informs about mechanical stresses, wall polymer breakdown or for regulated expansion growth (*Cheung and Wu, 2011*; *Lindner et al., 2012*). The precisely localized, ring-like Casparian strip can be used as a model for studying localized wall deposition (*Lee et al., 2013*). The *SGN3* LRR-RLK appears to be involved in both the formation of the CSD and the signaling of defective Casparian strips. Lack of *SGN3* causes an inability of CASP1-GFP patches to fuse into a contiguous band, without affecting the positioning of the patches along a median ring, nor the stability or functionality of the already established patches. SGN3 localization to the transversal and anticlinal, but not the periclinal membrane domains further increases the complexity of subdomains present at the endodermal plasma membrane. It embeds the forming CASP domain into a larger subdomain within which the narrower CSD is formed. Direct interaction of CASPs and SGN3 could take place early during CASP accumulation at the membrane and occur at the rims of forming CASP patches, possibly promoting patch growth and fusion. This can now be tested in future experiments. A second, currently unrelated role for *SGN3* is the induction of increased lignin

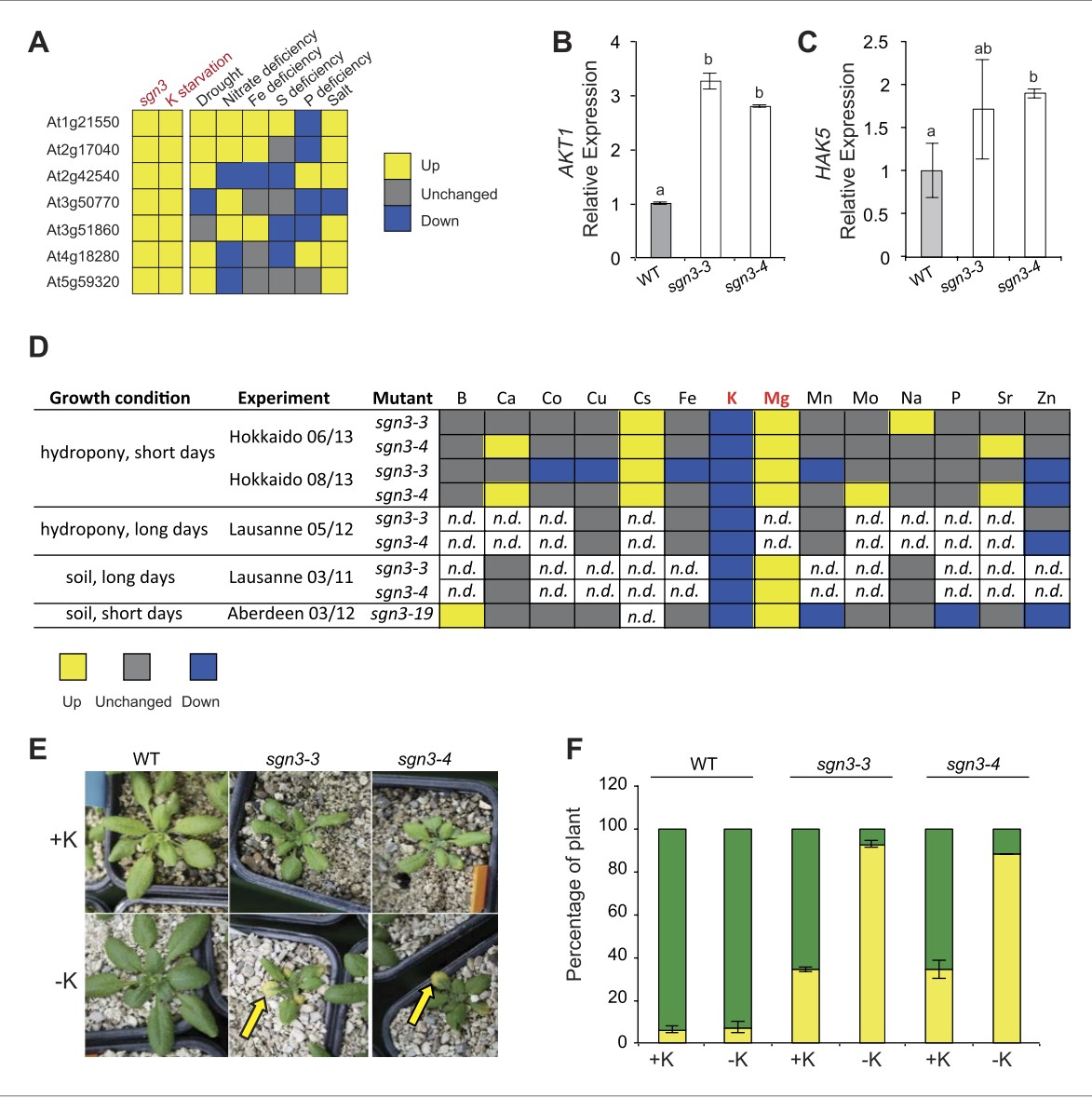

**Figure 5**. Potassium homeostasis is affected in *sgn3*. (**A**) Expression of genes upregulated in 4 week-old *sgn3* leaves. The seven genes presented here are the ones whose expression level was significantly increased in *sgn3* (p < 0.15). Those genes were investigated in Genevestigator for responses in leaves to nutritional stresses such as potassium starvation, drought, nitrate deficiency, Fe deficiency, S deficiency, P deficiency, and salt stress. Color-code indicates the effect of growth condition to their expression level (yellow up, gray unchanged, blue down). For numerical values see **Figure 5—source data 1**. (**B** and **C**) Quantitative RT-PCR analysis of *AKT1* (**B**) and *HAK5* (**C**) transcript levels in WT, *sgn3-3*, and *sgn3-4* roots (n = 3). Error bars = s.d, different letters indicate significant differences between genotypes, determined by analysis of variance (ANOVA) and Tukey test as *post hoc* analyses (p < 0.05). (**D**) Overview of ionomic analysis performed in *sgn3* leaves (*sgn3-3*, *sgn3-4*, and *sgn3-19* alleles) in three independent laboratories (Hokkaido, Lausanne, Aberdeen), using 2 growth systems (hydroponics or soil) and 2 day-length conditions (short or long days). Elements were determined by ICP-MS (Hokkaido, Lausanne 05/12 and Aberdeen) or ion chromatography (Lausanne 03/11). Color-code indicates significant changes of accumulation in *sgn3* mutants compared to WT (p < 0.05; yellow up, gray unchanged, blue down). For the numerical values, see **Figure 5—source data 2**. (**E** and **F**) Phenotype of 3-week-old WT, *sgn3-3*, and *sgn3-4* plants grown in potassium (K) deficiency. Plants were watered from germination with a nutritive solution with or without 1.5 mM $KNO_3$ (+K or −K). Representative picture are presented. Arrow indicates a chlorotic leave. (**F**) Occurrence of K deficiency phenotype determined as the percentage of plants displaying at least one yellow leaf (yellow) vs percentage of plants displaying only green leaves (green). Error bars = s.d.; data correspond to the mean of two independent experiments with a total of n ≥ 50 plants.

The following source data and figure supplements are available for figure 5:

**Source data 1**. Transcriptional differences between wild type and *sgn3* shoots.

*Figure 5. Continued on next page*

*Figure 5. Continued*

**Source data 2**. Overview of ionomic experiments.
**Figure supplement 1**. Ionomic comparision of WT and *sgn3* grown under low potassium.

and suberin formation in other Casparian strip mutants, such as *esb1* or *casp1 casp3*. We speculate that SGN3 acts as a kinase that reports CSD integrity, signaling an upregulation of lignin, and suberin production. A surveillance of Casparian strip integrity could be of great importance for unimpeded root function and would parallel the surveillance of tight junctions that has been reported for animal epithelia (*Balda and Matter, 2009*).

## SGN3 has redundant activities outside the endodermis

We could demonstrate that exclusive expression of SGN3 in the differentiating root endodermis is sufficient to rescue the overall growth defect of *sgn3*. This indicates that the endodermal function that we describe here is the only specific, non-redundant activity of this receptor in the plant. The earlier described function of *SGN3/GSO1* in the formation of an embryonic cuticle by contrast is fully redundant with that of its homolog *GSO2*. The dramatic cotyledonary phenotype of the double mutant is bound to have secondary consequences on seedling growth, making it very difficult to dissect primary from secondary effects. A recent investigation of root meristem defects of the double mutants, for example, could entirely be explained as secondary consequences of the primary defect in embryonic cuticle formation (*Racolta et al., 2014*). It is nevertheless important to speculate how the two functions of endodermal Casparian strip formation and protodermal cuticle formation could be explained by a common molecular activity. By genetic analysis, the *GSO1 GSO2* pair has been assigned to a signaling pathway that depends on an endosperm-expressed subtilase (ABNORMAL LEAF-SHAPE1, ALE1) (*Xing et al., 2013*). In a currently speculative scenario, subtilases might be involved in processing of peptide hormones that could act as ligands for SGN3/GSO1, promoting formation of a functional cuticle. Yet, the endodermal, lignified Casparian strip and the protodermal, cutin-based embryonic cuticle are two chemically different cell wall modifications that also show distinct modes and sites of deposition within the primary cell wall. These differences make it difficult to perceive a common denominator for SGN3 action in these two cell types. Moreover, our data suggest that a primary function of SGN3/GSO1 is the formation of a continuous band of CASP membrane proteins. No CASP homologs have been implicated in cuticle formation up-to-now, nor are there any indications for the presence of a highly scaffolded membrane protein domain in the epidermis/protodermis that would resemble the CSD. Another *SGN3/GSO1* function that we could establish is the promotion of suberin formation in response to a defective Casparian strip. Enhanced suberin formation in *esb1* appears to be entirely dependent on SGN3 activity, although the normal developmental progression of suberin formation is not affected in the mutant. Suberin and cutin are chemically closely related polymers whose monomers are formed by sets of homologous enzymes. A common role of *SGN3* might be to mediate a cross-talk between cell wall components, possibly regulating suberin production in endodermis and cutin production in epidermis/protodermis. Identification of the SGN3 ligands and downstream kinase substrates will be crucial for an understanding of the common mechanism that might underlie these apparently divergent functions.

## *sgn3* is the strongest and most specific Casparian strip mutant known to date

Regardless of the fact that *SGN3* has other roles in the plant, the *sgn3* single mutant defect reported here remains exquisitely specific to the endodermis. This is supported by the fact that it is only in the endodermis where *SGN3* expression is non-overlapping with its close homolog, *GSO2* (*Tsuwamoto et al., 2008*). Moreover, exclusive expression of *SGN3* in the endodermis is able to rescue the overall growth defects observed in aerial parts (this study). In addition to its specificity, the *sgn3* phenotype turns out to be strong and robust, that is, it displays penetration of tracer along the whole length of a seedling root and the phenotype is stable in the different growth conditions that we tested. Finally, *sgn3* does not show any of the enhanced suberin formation that has been observed in other mutants and which confounds interpretation of phenotypes. We therefore propose that the *sgn3* mutant is uniquely suited to obtain insights into the specific role of Casparian strips in plants. This mutant might be very

useful for investigating not only root nutrient and water uptake, but also for addressing the role of the endodermal barrier in pathogen resistance, hormonal transport, stress, or general growth responses.

## *sgn3* nutrient homeostatic defects are surprisingly specific

Our first analysis of the *sgn3* mutant has revealed an unexpected robustness of *Arabidopsis* growth towards a complete lack of its main root diffusion barrier. While still able to grow and complete its life cycle, the mutant nevertheless displays greatly exacerbated growth reductions to non-stressful changes in environmental conditions. This indicates the presence of homeostatic backup systems in the plant that are incompletely compensating for the absence of Casparian strips. Homeostasis of many elements is kept in the *sgn3* mutant, and it is intriguing that it is specifically potassium that shows a reproducibly strong decrease in concentration. Potassium (K) is the only macronutrient (N,P,K,S) that is not integrated into polymers and that remains highly mobile within the plants. It is being used as a counter ion in many transport process and is recycled back to the root along with the phloem sap. At the same time, concentrations of potassium required for plant growth are high and potassium often needs to be concentrated 10–100 fold over the concentration found in the soil (*Marschner and Marschner, 2012*; *Wang and Wu, 2013*). It therefore makes a lot of sense that potassium is an element that cannot be kept at sufficiently high levels in Casparian strip defective plants. We propose that potassium located in the stelar apoplast is continuously lost to the outer cell layers and the soil in the absence of functional Casparian strips. Upregulation of potassium influx carriers in cortex/epidermis, as we have observed in *sgn3* roots, would certainly be able to recuperate large parts of the potassium being lost. However, a chronic condition of very low external potassium would lead to a breakdown in this transporter-mediated 'potassium recycling' and eventually cause the deficiency phenotypes that we observe. From this scenario, it follows that there should be a higher rate of potassium flux out-of and into the root stele in a *sgn3* mutant and that potassium homeostasis in the mutant is extremely dependent on increased activity of potassium transporters. We are currently working on substantiating this 'potassium recycling' model by specific interference with and visualization of potassium transporters in the mutant. Further analysis of *sgn3* will be instructive for a better understanding of root potassium transport in general and this analysis should also be extended to other macronutrients in the future.

What could be the reasons for the lack of effects on many other elements? It can be expected that similar compensatory responses as seen for potassium exist for other elements. If these elements need to be less concentrated over their external concentration—or if they are less mobile within the plant—the compensatory responses might be sufficient for maintaining homeostasis in the mutant. Interestingly, we found an increase in the levels of the divalent cation magnesium, while concentrations of the divalent calcium, for example, were not affected in *sgn3* mutants. This could be explained by a lower concentration gradient between the xylem sap and the external environment and a lower mobility of Calcium within the cell wall space (*Clarkson, 1984*; *White and Broadley, 2003*). Nevertheless, calcium concentration within the symplast is kept extremely low for cellular signaling purposes. Therefore, calcium delivery to aerial tissues is thought to occur either entirely through the apoplast–using the few apoplastic 'bypasses' of the endodermis at the root tip or at lateral root emergence sites (*White, 2001*)—or to employ a very short symplastic route passing exclusively through endodermal cells (*Clarkson, 1984*). In both scenarios, it is surprising that the opening of a massive apoplastic bypass in the *sgn3* mutant has no effect on calcium homeostasis, while it did affect levels of magnesium. Possibly, magnesium is more mobile in the apoplast than Calcium and undergoes increased apoplastic transport into the transpiration stream in *sgn3* (*Thibault and Rinaudo, 1985*). Identifying conditions in which calcium homeostasis also breaks down in *sgn3* would greatly advance our understanding of the calcium uptake mechanisms in plants.

The *sgn3* mutant represents a powerful new tool to better understand root uptake and transport mechanism of most other plant nutrients as well. Our initial analysis of the *sgn3* mutant challenges the current notions of the role of the Casparian strip as a required barrier for most or all elements. The ability to selectively disrupt this strict endodermal barrier will allow for direct investigations of other mechanisms that maintain and buffer ion homeostasis. Unmodified cell walls, by their properties as ion exchange matrices, in conjunction with vacuolar storage of elements, can play pivotal roles in maintaining homeostasis of many elements. Cytoplasmic magnesium levels, for example, might well remain unaltered in *sgn3*, the increased magnesium levels being entirely absorbed by the vacuole. Having taken away the 'first lock' of the Casparian strip now puts the focus on the existence of 'second locks' that can be studied using the *sgn3* mutant as a sensitized background.

# Materials and methods

## Plant material

*Arabidopsis thaliana* ecotype Columbia was used for all experiments. For details of knockout mutants, see *Table 1*.

## Vector construction and transgenic lines

For generation of expression constructs, Gateway Cloning Technology (Invitrogen, Carlsbad, CA) or standard molecular biology procedures were used. *pESB1::ESB1-mCherry* plasmid (*Hosmani et al., 2013*) was transformed in *sgn3-3* background. *pSGN3* (5583 bp before ATG)::*SGN3 gDNA-mVenus* was cloned by integrating a 9.4-kb genomic fragment including the intron and the 5'UTR into a Basta-resistance pGREENII vector containing *mVenus*. *pCASP1::NLS-GFP-GUS* and *pCASP4::NLS-GFP-GUS* reporters were cloned into a Basta-resistance pGREENII vector and contain 1048 bp and 695 bp promoter fragments, respectively. *pCASP1::SGN3 cDNA-GFP* was cloned with Gateway using a 1207 promoter fragment followed by *SGN3* cDNA without the unique intron and *GFP*. *pCASP1* (1207 bp)::*PDR6 gDNA-Venus-4G* (4 glycine extension) was cloned using Gateway in a pB7m34 GW,3 expression vector (http://gateway.psb.ugent.be). *pCASP1* (1207 bp)::*mCherry-SYP122* was constructed with Gateway by cloning the *AtSYP122* (*At3g52400*) cDNA into pH7m42 GW,3. *pCASP1::CASP2-GFP-4G*, *pCASP1::CASP3-GFP-4G*, *pCASP1::CASP4-mCherry* and *pCASP1::CASP5-GFP-4G* were cloned with Gateway using the pB7m34 GW,3 expression clone. Transgenic plants were generated by introduction of the plant expression constructs into an *Agrobacterium tumefaciens* strain GV3101 and transformation was done by floral dipping (*Clough and Bent, 1998*).

## Growth conditions

For in vitro assays, plants were germinated on 0.5 MS (Murashige and Skoog) agar plates after 2 days in dark at 4°C. Seedlings were grown vertically in growth chambers at 22°C, under long days (16-hr light/8-hr dark), 100 μE light, and were used at 5 days after shift to room temperature. For microarray, transpiration, hydraulic conductivity, and ionomic analysis in Lausanne (May 2012) assays, plant were germinated on hydroponic conditions after 2 days in dark at 4°C. Nutrient solution was changed weekly and contained 0.5 mM $KNO_3$, 0.25 mM $Ca(NO_3)_2$, 1 mM $KH_2PO_4$, 1 mM $MgSO_4$, $FeNH_4$-EDTA 0.1 mM, KCl 50 μM, $H_3BO_3$ 30 μM, $MnSO_4$ 5 μM, $ZnSO_4$ 1 μM, $CuSO_4$ 1 μM, $(NH_4)_6Mo_7O_{24}$ 0.1 μM. Light/dark cycle was 16 hr/8 hr, 23°C/18°C, 100 μE light.

For xylem sap and guttation assays, plants were germinated after 2 days in dark at 4°C on soil in short day condition and irrigated with water. Light/dark cycle was 10 hr/14 hr, 22°C/18°C, 100 μE light.

For phenotypic assays at different light intensities (30, 90, 150, or 300 μE), temperatures (17, 21, or 25°C), or day lengths (8, 16 or 24 hr), plants were germinated after 2 days in dark at 4°C on soil and irrigated with water. The standard condition was: light/dark cycle 16 hr/8 hr, 21°C/19°C with light intensity of 150 μE. Temperature difference between day and night was always kept at 2°C.

**Table 1.** Details of knockout mutants

| Gene number | Accession | Mutant number | Mutant name | Genotyping primer sequence | References |
|---|---|---|---|---|---|
| *AT4G20140* | Col-0 | SALK_043282 | *sgn3-3* | LP: ATTCTACGAGCCTTCCCATTC<br>RP: CGCAGTGAACACAGTGAGATC | Present work |
| *AT4G20140* | Col-0 | SALK_064029 | *sgn3-4 or gso1-1* | LP: CTCGGCTCCCTCGTTAATATC<br>RP: GTTACCTAAACTGGCGGGAAG | *Tsuwamoto et al. (2008)*<br>The Plant Journal |
| *AT4G20140* | Col-0 | SALK_103965 | *sgn3-19* | LP: TCCATTATGTGGTTCGAGCTC<br>RP: CTTGTAAACCTTCCCAGAGCC | Present work |
| *AT2G28670* | Col-0 | n.a. | *esb1-1* | n.a. | *Baxter I et al. (2009)* PLOS Genet,<br>*Lahner B et al. (2003)* Nat Biotechnol |
| *AT2G36100* | Col-0 | SAIL_265_H05 | *casp1-1* | LP: GCGTTTCAGTACGTCCCTTC<br>RP: CACGTGAGGGAAGTGAGTCTC | *Roppolo et al. (2011)* Nature |
| *AT2G27370* | Col-0 | SALK_011092 | *casp3-1* | LP: GACTCTTCCTTTCTTCACTC<br>RP: GACCAACACAACCGTACGAAC | *Roppolo et al. (2011)* Nature |

For ionomic assays plants were grown in different laboratories as indicated. For the Hokkaido experiments plants were grown hydroponically as described in *Takano et al. (2001)*, with slight modifications. Environmental parameters in a growth chamber are as follows: 10-hr/14-hr light/dark cycle, 22°C under fluorescent lamps, 70% humidity. The seeds were sown on rockwool and grown supplied with hydroponic media (*Fujiwara et al., 1992*) supplemented with 50 μM Fe-EDTA. After 15 days, the plants were additionally supplied with 3 μM CsCl and 10 μM $SrCl_2$. The media were changed twice a week. For the Aberdeen experiment plants were grown in short day conditions, in Bulrush multi-purpose compost (http://www.bulrush.co.uk/retail-range/all-purpose-composts.html) spiked with various elements as detailed in *Lahner et al. (2003)*, and bottom watered using ¼ Hoagland solution containing Fe as 10 μM Fe-HBED as described in *Baxter et al. (2008)*. 1–2 adult rosette leaves were harvested 36 days after planting. For the Lausanne experiment in soil (March 2011), plants were grown in long days 16-hr/8-hr light/dark, 23°C/19°C, and leaves were harvested after 5 weeks.

For potassium deficiency assay plants were germinated after 2 days in dark at 4°C on a poor gravel-like substrate (OIL DRI US-Special) watered with a ½ Hoagland based solution containing : 1.5 mM $NH_4NO_3$ or 1.5 mM $KNO_3$ (−K or +K).

## Confocal microscopy

Confocal laser scanning microscopy experiments were performed either on a Leica SP2, a Zeiss LSM 700, a Zeiss LSM 710, or a Zeiss LSM 710 NLO 2-Photon microscope. Excitation and detection windows were set as follows: Leica SP2: GFP 488 nm, 500–600 nm; propidium iodide 488 nm, 600–700. Zeiss LSM 700: GFP/mVenus 488 nm, 490–555 nm; mCherry/propidium iodide 555 nm, SP 640. LSM 710: GFP/mVenus 488 nm, 495–554 nm; mCherry/propidium iodide 561 nm, 573–681 nm. Zeiss LSM 710 NLO 2-Photon: GFP/mVenus, 960 nm, 500–550 nm (NDD). For visualization of the apoplastic barrier, seedlings were incubated in the dark for 10 min in a fresh solution of 15 mM (10 mg/ml) Propidium Iodide (PI) and rinsed two times in water (*Alassimone et al., 2010*, *Naseer et al., 2012*). For quantification, 'onset of elongation' was defined as the point where an endodermal cell in a median, longitudinal section reached a length more than twice its width. From this point, cells in the file were counted until the PI signal was blocked in the endodermal cells. For plants grown in hydroponic and in soil, roots of 13 and 14 days were used, respectively.

Casparian strip autofluorescence after clearing was visualized as described (*Alassimone et al., 2010*, *Naseer et al., 2012*).

Fluorescence recovery after photobleaching (FRAP) was performed with 5-day-old seedlings and imaged with a Leica SP2 confocal microscope as described (*Roppolo et al., 2011*).

Plasmolysis was induced by incubating 5-day-old seedlings for 1 hr in 0.8 M mannitol and then mounted in the same solution. Plasmolyzed cells were imaged at 15 cells after the onset of *CASP1* expression.

For time-lapse imaging of *CASP1::CASP1-GFP* in Col-0 and *sgn3-3*, 5-day-old seedlings were into a Lab-Tek II chambered coverglass (Nunc) and covered with a small block of agar to prevent drying. Subsequently, slides were mounted and imaged on an upright Zeiss LSM710 NLO confocal microscope. Excitation was provided by a Ti-Sapphire Chameleon II Ultra (Coherent) with 960 nm and fluorescence was detected using non-descanned detection (NDD) between 500 and 550 nm. Image stacks were taken every 15 min for a total time of 16 hr. Confocal images were analyzed and contrast and brightness were adjusted with the FIJI package (http://fiji.sc/Fiji) and Adobe Photoshop CS5.

For the percentage quantification of Casparian strips, autofluorescence after clearing was performed on ten seedlings for each genotype investigated. For each individual seedling, Casparian strips were imaged by confocal microscopy by doing one z-stack and measurements were done using ImageJ. The percentage of Casparian strip was the portion where autofluorescence was visible along a line showing Casparian strip signal. A minimum of 3.5 mm of Casparian strip was quantified in each genotype.

## Suberin lamellae analysis

Suberin lamella was observed in 5-day-old roots after Fluorol Yellow staining as described in *Naseer et al. (2012)*. Seedlings were incubated in Fluorol Yellow 088 (0.01% wt/vol, lactic acid) at 70°C for 30 min, rinsed with water, and counterstained with aniline blue (0.5% wt/vol, water) at RT for 30 min in darkness, washed, mounted on slides with glycerol and observed with epifluorescence microscope. Suberin pattern were observed and counted from the hypocotyl junction to the onset of endodermal cell elongation. Three distinct patterns were considered: continuous suberin lamellae, discontinuous

suberin lamellae (corresponding to the area where suberin lamellae establish), and non- suberized (corresponding to the young part of the root). Experiments were repeated at least four times.

## Water transport assay

### Transpiration

For transpiration assays, 3-week-old plants grown in hydroponic conditions were transferred to airtight containers isolating the root and nutritive solution from the air while plant shoots remain out of the container (potometer). Transfer to potometer was performed 24 hr prior to transpiration measurement. Transpiration was determined as the loss of weight measured every 2 hr during 14 hr and was expressed relatively to the plant's leaf area. Leaf area was measured for each individual plant, with ImageJ from pictures of detached leaves at the end of the transpiration assay.

### Root pressure

Plants were grown 6 weeks in short day condition and decapitated with scalpel at the hypocotyl junction. The decapitated hypocotyl was immediately introduced into a 100 µl microcapillary. Root pressure was determined as the volume of xylem sap collected within 30 min from decapitated plants.

### Guttation

Plants were grown 6 weeks in short day condition. To induce guttation, plants were transferred to 15°C in the dark for 12 hr, covered with a lid to maintain high humidity. The liquid was harvested from 15 individual plants for each genotype, using a micropipette.

### Hydrostatic hydraulic conductivity

Measurements of root hydrostatic hydraulic conductivity ($Lp_{r-h}$) were performed as described previously (*Javot et al., 2003*; *Postaire et al., 2010*).

## Ionomic analysis

For ionomic assays plants were grown in different laboratories as indicated. For the Hokkaido and Lausanne experiments (hydroponic conditions), the shoots of plants were harvested, dried in an air incubator at 60°C for more than 60 hr, and the dry weights were measured. The tissues were digested with 3 ml of 61% $HNO_3$ (for boron determination; Wako Pure Chemicals, Osaka, Japan) in a tube at 110°C in a DigiPREP apparatus (SCP Science, Quebec, Canada) until complete dryness. The residues were dissolved in 10 ml of 2% $HNO_3$ and analyzed for elements by inductively coupled plasma mass spectrometry (ICP-MS) (ELAN, DRC-e; Perkin–Elmer, Waltham, MA, USA).

For Aberdeen assays, the leaves were cleaned by rinsing with ultrapure water and placed into Pyrex digestion tubes. Samples were dried in an oven at 88°C for 20 hr. After cooling, seven reference samples from each planted block were weighed. The samples together with blank controls were digested with 0.90 ml concentrated nitric acid (Baker Instra-Analyzed; Avantor Performance Materials) and diluted to 10.0 ml with ultrapure water (18.2 MΩcm). The internal standard Indium (In) was added to the acid prior to digestion for monitoring technical errors and plasma stability in the ICP-MS instrument. After samples and controls were prepared, elemental analysis was performed with an ICP-MS (NexION 300D; PerkinElmer) coupled to Apex desolvation system and SC-4 DX autosampler (Elemental Scientific Inc., Omaha, NE, USA), monitoring these elements: B, Na, Mg, P, K, Ca, Mn, Fe, Co, Cu, Zn, Sr, and Mo. All samples were normalized to calculated weights, as determined with a heuristic algorithm using the best-measured elements, the weights of the seven weighed samples and the solution concentrations (*Lahner et al., 2003*), detailed at www.ionomicshub.org.

For ionic chromatography, for each sample one rosette leaf of a 5-week-old plant was harvested, put in 30 ml Nanopure water and incubated at 80°C for 1 hr. Each sample was then filtered with a 0.45-µm non-pyrogenic sterile filter (Sarstedt, Germany). The samples were then diluted to reach a final concentration of 0.83 mg fresh weight per 1 ml. Ion concentrations were obtained with an ion chromatography system (ICS-1100/2100, Dionex-Thermo Fischer).

## Expression analysis

### Microarray analysis

WT and *sgn3-3* plants were grown in hydroponic conditions and material was harvested 24 days post-stratification. Roots and rosettes were collected separately, and each sample is a mix of three

plants. Each type of sample was harvested in triplicate. In these growth conditions, sgn3-3 was nearly indistinguishable from wild type. Total RNAs were isolated and purified with RNeasy Plant Mini Kit (Qiagen, Venlo, The Netherlands). RNA quantities were assessed with a NanoDropND-1000 spectrophotometer and RNA qualities with an RNA 6000 NanoChips with the Agilent 2100 Bioanalyzer (Agilent, Palo Alto, USA). For each sample, 300 ng of total RNA were amplified using the message amp II enhanced (AM1791, Ambion) kit. 12.5 μg of the resulting biotin-labeled cRNA was chemically fragmented. Affymetrix ATH1 arrays (Affymetrix, Santa Clara, CA) were hybridized with 11 μg of fragmented target at 45°C for 17 hr washed and stained according to the protocol described in Affymetrix GeneChip Expression Analysis Manual (Fluidics protocol FS450_0007). The arrays were scanned using the GeneChip Scanner 3000 7 G (Affymetrix). All statistical analyses were performed using the free high-level interpreted statistical language R and various Bioconductor packages (http://www. Bioconductor.org). Hybridization quality was assessed using the Expression Console software (Affymetrix). Normalized expression signals were calculated from Affymetrix CEL files using RMA normalization method. Differential hybridized features were identified using Bioconductor package 'limma' that implements linear models for microarray data (*Smyth, 2004*). The p-values were adjusted for multiple testing with Benjamini and Hochberg's method to control the false discovery rate (FDR) (*Benjamini and Hochberg, 1995*). Genevestigator was used to analyze the effect of nutritional stress on the expression of genes deregulated in sgn3 (*Hruz et al., 2008*).

## Real time Q-PCR

Total RNA was extracted using TRIzol reagent (Invitrogen) and purified using RNeasy MinElute Cleanup Kit (Qiagen) after DNase treatment (Qiagen). The integrity of DNA-free RNA was verified by PCR, agarose gel electrophresis, and nanodrop.

An equal amount of total RNA (3 μg) was used for RT with anchored oligo(dT23). Real time Q-PCR was performed using a Stratagene MxPro 3005P Real-Time PCR System (Stratagene, La Jolla, CA). Three biological replicates performed in three technical replicates were analyzed for each sample. The specificity of each amplification product was verified by DNA melting curve analysis and gel electrophoresis of the amplified products. Relative transcript levels (RTL) were calculated relative to the transcript level of the reference gene *EF1α*, as follows: RTL = 1000 × $2^{-\Delta Ct}$ with ΔCt, the change in cycle threshold between the target gene and the reference gene. The following oligonucleotides were used *EF1α* (*At1g07920, At1g07930, At1g07940*) 5'-GTCGATTCTGGAAAGTCGACC-3' and 5'-AATGTCAATGG TGATACCACGC-3'; *CASP1* (*At2g36100*) 5'-AGAGAGGTTTGGCTATATT-3' and 5'-CTACGGCTATCA CAAAGTA-3'; *CASP4* (*At5g06200*) 5'-AGACTTCTCTTGCTTGTTCT-3' and 5'-GACAGAAATCTCC AAACTG-3'; *AKT1* (*At2G26650*) 5'-TCTAAATTGTGTTCTTCTTCTGTTGGA-3' and 5'-CCTTCCGCG TCTCTGCAA-3'; *HAK5* (*At4g13420*) 5'-CGAGACGGACAAAGAAGAGGAACC-3' and 5'- CACGA CCCTTCCCGACCTAATCT-3'.

## Statistical analysis

All statistical analyses were done in the R environment (http://www.R-project.org/). Binary comparisons between wild type and sgn3 were performed using Student's *t*-test. Multiple comparisons between wild type, sgn3-3, and sgn3-4 mutants, one-way ANOVA was performed, and Tukey's test was subsequently used as a multiple comparison procedure.

## SGN3 domain identification

The different features were predicted using the following web resources. Signal peptide, Leucine-rich repeats, transmembrane domain, and kinase domain: http://www.uniprot.org/uniprot/C0LGQ5; LRR N-terminal domain: http://pfam.sanger.ac.uk/search/sequence; kinase sub-domains were defined with reference to *Hanks and Hunter (1995)* and by aligning to the BRI1 kinase domain (*Vert et al., 2005*). Borders between kinase sub-domains were set manually by comparison of amino acids in the respective border regions for PKA C alpha and BRI1. In case of conflict, the BRI1 annotation was considered as more meaningful because of its plant origin.

## Acknowledgements

We thank the Genomic Technologies Facility (GTF) and the Central Imaging Facility (CIF) of the University of Lausanne for expert technical support. We thank Valérie Dénervaud Tendon, Guillaume Germion, Deborah Mühlemann, and Kayo Konishi for technical assistance and John Danku and Véronique Vacchina for ICP-MS analysis. This work was funded by grants from the Swiss National

Science Foundation (SNSF), the European Research Council (ERC) to NG and a Human Frontiers Science Program (HFSP) grant to JT and NG. GL and CM were supported by the Agropolis foundation (Rhizopolis) and the Agence Nationale de la Recherche (HydroRoot; ANR-11-BSV6-018). MB was supported by a EMBO long-term postdoctoral fellowship, JEMV by a Marie Curie IEF fellowship and TK by the Japan Society for the Promotion of Sciences (JSPS).

## Additional information

### Funding

| Funder | Author |
|---|---|
| European Research Council | Alexandre Pfister, Marie Barberon, Julien Alassimone, Lothar Kalmbach, Yuree Lee |
| Swiss National Science Foundation | Alexandre Pfister, Marie Barberon, Julien Alassimone, Lothar Kalmbach, Yuree Lee |

The funders had no role in study design, data collection and interpretation, or the decision to submit the work for publication.

### Author contributions

AP, MB, JA, LK, YL, JEMV, Conception and design, Acquisition of data, Analysis and interpretation of data, Drafting or revising the article; MY, Acquisition of data, Contributed unpublished essential data or reagents; GL, Acquisition of data, Analysis and interpretation of data; CM, JT, DES, NG, Conception and design, Analysis and interpretation of data, Drafting or revising the article; TK, DR, Conception and design, Acquisition of data, Analysis and interpretation of data

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
