## [Decision Letter]

Thank you for sending your work entitled “A receptor-like kinase mutant with absent endodermal diffusion barrier displays selective nutrient homeostasis defects” for consideration at *eLife.* Your article has been favorably evaluated by Detlef Weigel (Senior editor) and 3 reviewers, one of whom is a member of our Board of Reviewing Editors.

The Reviewing editor and the other reviewers discussed their comments before we reached this decision, and the Reviewing editor has assembled the following comments to help you prepare a revised submission.

In this interesting study the authors identify a receptor-like kinase gene, SGN3/GASSHO1, that is essential for Casparian strip formation. Through a series of elegant cell biology and genetics experiment they demonstrate that SGN3 is an upstream signaling component that plays a role in the progression of stable CASP islands during Casparian strip formation. *sgn3* is the first mutant with a permanently discontinuous Casparian strip and this enabled the authors to demonstrate the functional significance of the Casparian strip in controlling root pressure, hydroponic conductivity and mineral nutrient homeostasis. The reviewers' main concerns relate to the claims about mineral nutrient phenotypes and the following revisions are required.

1) The authors place tremendous emphasis on potassium (K) and conclude that this is the only macronutrient that is greatly affected in the mutant. However, data for nitrogen (N) or sulfur (S) are not included while limited phosphorus (P) data suggest an approximately 60% reduction in P accumulation when plants are grown in soil. Thus, it is possible that both macro-nutrients K and P are affected. This requires further evaluation and additional data for K and P levels in soil grown plants should be included. In addition, the lack of evaluation of N and S should be acknowledged and the claims about the specificity of the K phenotype should be removed.

2) The low and high K experiment (Figure 5) shows a difference in plant growth which is attributed to differences in K levels. In order to support this, the leaf K levels of these plants should be presented. (The yellow leaf phenotype could arise from differences in nitrate, which appear to differ in the low and high K treatment as KNO3 was omitted from the low K treatment.)

3) The iron levels are higher than potassium levels in the hydroponics experiments (Figure 5–figure supplement 2). It would be helpful to readers to add a comment on this observation.

4) The authors' statements, that the lack of effects on most nutrients counters the present (“text book”) understanding of plant nutrition, do not appear to be up-to-date, especially for micronutrients. It is known that high-affinity (secondary) active transporters and endodermal exporters are important for micronutrient uptake and ultimate transfer to the xylem (and phloem) and a somewhat leaky Casparian strip should have either little or perhaps only small secondary effects on nutrient accumulation, especially for those using (secondary) active transport. It is likely that out-of-date text books can be found, but the old model referenced by the authors does not conform to the present understanding of the electrical and energetic properties of many nutrient transporters. These over-statements should be removed.

5) The nutrient content of the media and fertilizers used are important for the conclusions and should be provided fully for all experiments in the Methods section.

6) The text of the manuscript should be edited to make it more concise.

7) As shown in Figure 5, the *sgn3* transcript data are similar to drought and salt responses. The latter could be further evaluated by running a salt stress experiment and assessing plant growth and ion content of roots and leaves. As salt stress occurs at high millimolar NaCl concentrations, by-pass transport is likely to be more relevant in the *sgn3* mutant, similar to NaCl transfer in natural rice variants. This experiment is not essential but the reviewers felt that this would strengthen the case for ion-related phenotypes in *sgn3*. The experiment is not difficult and could be run in parallel with the experiment that further evaluates K and P levels in soil/gravel grown plants as requested in point 1.

[Editors' note: further revisions were requested prior to acceptance, as described below.]

Thank you for resubmitting your work entitled “A receptor-like kinase mutant with absent endodermal diffusion barrier displays selective nutrient homeostasis defects” for further consideration at *eLife.* The manuscript has been improved but there are some remaining issues that need to be addressed before acceptance, as outlined below:

1) You elected to focus on changes in potassium in *sgn3*. At a minimum, leaf potassium levels should be provided for the low and high K experiment (soil-like substrate grown plants) shown in Figure 5. This is needed to support the claim that the growth and leaf chlorosis phenotypes observed in *sgn3* are the result of a reduction in K levels. Analysis of other nutrients in soil grown plants is optional, but it could provide useful evidence to support other nutrient defects in *sgn3*.

2) If changes in nutrients other than potassium are not confirmed, it might be more accurate to alter the title to reflect potassium rather than selective nutrients.

---

## [Author Response]

*1) The authors place tremendous emphasis on potassium (K) and conclude that this is the only macronutrient that is greatly affected in the mutant. However, data for nitrogen (N) or sulfur (S) are not included while limited phosphorus (P) data suggest an approximately 60% reduction in P accumulation when plants are grown in soil. Thus, it is possible that both macro-nutrients K and P are affected. This requires further evaluation and additional data for K and P levels in soil grown plants should be included. In addition, the lack of evaluation of N and S should be acknowledged and the claims about the specificity of the K phenotype should be removed*.

We agree that our phrasing suggested that specifically K is affected among the macronutrients, which we simply cannot claim due to lack of data. We have reformulated the sentence in the Discussion. We also agree that it would be worthwhile to investigate also N, P, and S. However, we were very careful in this first paper to only present robust changes that reproducibly occurred in all measurements, from us and in the labs of our collaborators. Therefore, we would not like to present possible changes in other macronutrients that have only be found in one experimental set in our laboratory, which is, however, the only thing we would be able to deliver in a reasonable time-scale for this revision. We would therefore prefer not to add and discuss other macronutrient data in this work. As suggested, we have added a sentence in the discussion that acknowledges the lack of macronutrient data.

*2) The low and high K experiment (*Figure 5*) shows a difference in plant growth which is attributed to differences in K levels. In order to support this, the leaf K levels of these plants should be presented. (The yellow leaf phenotype could arise from differences in nitrate, which appear to differ in the low and high K treatment as KNO3 was omitted from the low K treatment*.*)*

We thank the reviewers for this comment. We actually incorrectly described this in our Methods. We are sorry about this. We did not simply omit KNO3, but replaced it with NH4NO3, which excludes the possibility that the yellowing arises from deficiency in Nitrate.

*3) The iron levels are higher than potassium levels in the hydroponics experiments (Figure 5–figure supplement 2). It would be helpful to readers to add a comment on this observation*.

We thank the reviewers for this comment. We had a unit problem in our data, caused by formatting problems (microgram to milligram). This has been corrected now. We are sorry for this confusion.

*4) The authors' statements, that the lack of effects on most nutrients counters the present (“text book”) understanding of plant nutrition, do not appear to be up-to-date, especially for micronutrients. It is known that high-affinity (secondary) active transporters and endodermal exporters are important for micronutrient uptake and ultimate transfer to the xylem (and phloem) and a somewhat leaky Casparian strip should have either little or perhaps only small secondary effects on nutrient accumulation, especially for those using (secondary) active transport. It is likely that out-of-date text books can be found, but the old model referenced by the authors does not conform to the present understanding of the electrical and energetic properties of many nutrient transporters. These over-statements should be removed*.

We have removed the “textbook” mention in the Discussion and elsewhere, which we agree might be seen as an overstatement. We maintain, however, that our current understanding of the Casparian strip is that of a fundamental diffusion barrier in the apoplast of the root, whose absence should either cause excessive influx of elements into the xylem or a loss of accumulated elements from it. To our knowledge there is no data or predictive model at present that would allow us to understand why a given element is increased, decreased or unaltered in the absence of an endodermal barrier. If the editor or reviewers know of a current textbook, review or original paper that addresses this question, we would be very grateful for a reference. We are also very much aware of the manifold ways that plant cells use to actively transport nutrients into or out-of the cell. Yet, we fail to see how any transporter, channel or pump, driven by ATP, using proton or other gradients, in symport or antiport, electrogenic or not, would be able to control diffusion of ions from the apoplastic space of the cortex/soil to the apoplastic space of the xylem if there is no diffusion barrier. Transport proteins can only control what goes into and out-of the symplast. What the *sgn3* mutations do is to create an *apoplastic bypass* that will allow gradients to dissipate between the stelar and cortical apoplast generated by the activity of transporters. The importance of transport into and out-of the symplast has been heavily investigated in recent decades through the identification and characterization of a plethora of transport proteins. During this period, not much attention has been paid to the relevance of the Casparian strip, because of a lack of tools and the models about its function have essentially remained unaltered for decades because of this.

*5) The nutrient content of the media and fertilizers used are important for the conclusions and should be provided fully for all experiments in the Methods section*.

We have provided this information.

*6) The text of the manuscript should be edited to make it more concise*.

We agree. We have been able to shorten the main text considerably. In order to focus the story, we also eliminated the part that described CASP4 as a downstream target, which is of unclear functional relevance and really doesn’t contribute to the story.

*7) As shown in*
Figure 5*, the* sgn3 *transcript data are similar to drought and salt responses. The latter could be further evaluated by running a salt stress experiment and assessing plant growth and ion content of roots and leaves. As salt stress occurs at high millimolar NaCl concentrations, by-pass transport is likely to be more relevant in the* sgn3 *mutant, similar to NaCl transfer in natural rice variants. This experiment is not essential but the reviewers felt that this would strengthen the case for ion-related phenotypes in* sgn3*. The experiment is not difficult and could be run in parallel with the experiment that further evaluates K and P levels in soil/gravel grown plants as requested in point 1.*

I think one should not over-interpret the very weak changes in the transcriptional profile that we observed and that we would not have presented if they weren’t confirmed by qPCR data that indicates upregulation of AKT1 and HAK5. Nevertheless, we have tested repeatedly for salt and drought stress, with variable results. In the end we concluded that there certainly is no strong, reproducible drought or salt sensitivity of *sgn3* in our hands, although colleagues of ours might have found some degree of salt sensitivity of *sgn3*. In this context, we would prefer not to present data on drought or salt sensitivity. We would like to point out that this paper is the result of years of screening of many different nutrient stress conditions and that we only present in this work the data that we feel is the most instructive, robust, and reproducible.

[Editors' note: further revisions were requested prior to acceptance, as described below.]

*Thank you for resubmitting your work entitled “A receptor-like kinase mutant with absent endodermal diffusion barrier displays selective nutrient homeostasis defects” for further consideration at* eLife*. The manuscript has been improved but there are some remaining issues that need to be addressed before acceptance, as outlined below:*

*1) You elected to focus on changes in potassium in* sgn3*. At a minimum, leaf potassium levels should be provided for the low and high K experiment (soil-like substrate grown plants) shown in*
Figure 5*. This is needed to support the claim that the growth and leaf chlorosis phenotypes observed in* sgn3 *are the result of a reduction in K levels. Analysis of other nutrients in soil grown plants is optional, but it could provide useful evidence to support other nutrient defects in* sgn3*.*

We have done an entire elemental profiling experiment, including the measurement of potassium levels. *sgn3* maintains lower potassium levels than wild-type under –K conditions and no other changes in the elemental profile are observed, beyond those already reported in the numerous +K condition experiments provided in this manuscript.

We added the following sentence to the last paragraph of the Results section of the manuscript: “Consistently, wild-type also maintained higher potassium concentration than *sgn3* under these conditions (Figure 5–figure supplement 3).”

*2) If changes in nutrients other than potassium are not confirmed, it might be more accurate to alter the title to reflect potassium rather than selective nutrients*.

They are confirmed: with the two additional ionomic comparisons provided in the revision, we now provide seven elemental comparisons between *sgn3* and wt in total. They all show the low potassium, high magnesium profile that we discuss as being the most robust changes. We therefore would like to leave the title as it is.